# ARPC1B Is Associated with Lethal Prostate Cancer and Its Inhibition Decreases Cell Invasion and Migration In Vitro

**DOI:** 10.3390/ijms23031476

**Published:** 2022-01-27

**Authors:** Yaser Gamallat, Hend Zaaluk, Ealia Khosh Kish, Ramy Abdelsalam, Konstantinos Liosis, Sunita Ghosh, Tarek A. Bismar

**Affiliations:** 1Department of Pathology and Laboratory Medicine, Cumming School of Medicine, University of Calgary, Calgary, AB T2V 1P9, Canada; yaser.gamallat@ucalgary.ca (Y.G.); zaaluk@ucalgary.ca (H.Z.); ealia.khoshkish@ucalgary.ca (E.K.K.); ramyabdelsalam666@gmail.com (R.A.); 2Alberta Precision Laboratories, Calgary, AB T2V 1P9, Canada; 3Department of Computer Science, University of Calgary, Calgary, AB T2V 1P9, Canada; konstantinos.liosis@ucalgary.ca; 4Department of Medical Oncology, Faculty of Medicine and Dentistry, University of Alberta, Edmonton, AB T6G 2R7, Canada; sunita.ghosh@albertahealthservices.ca; 5Departments of Mathematical and Statistical Sciences, University of Alberta, Calgary, AB T2V 1P9, Canada; 6Departments of Oncology, Biochemistry and Molecular Biology, Calgary, AB T2V 1P9, Canada; 7Tom Baker Cancer Center, Arnie Charbonneau Cancer Institute, Calgary, AB T2V 1P9, Canada

**Keywords:** ARPC1B, prostate cancer, immigration, invasion, prognosis, ERG, PTEN

## Abstract

ARPC1B (Actin Related Protein 2/3 Complex Subunit 1B) has been found to be involved in platelet abnormalities of immune-mediated inflammatory disease and eosinophilia. However, its role in prostate cancer (PCa) has not been established. We characterized the role of ARPC1B in PCa invasion and metastasis and investigated its prognosis using in vitro cellular models and PCa clinical data. Higher immunohistochemistry (IHC) expressions of ARPC1B were observed in localized and castrate resistant PCa (CRPC) vs. benign prostate tissue (*p* < 0.01). Additionally, 47% of patients with grade group 5 (GG) showed high ARPC1B expression vs. other GG patients. Assessing ARPC1B expression in association with two of the common genetic aberrations in PCa (ERG and PTEN) showed significant association to overall and cause-specific survival for combined assessment of ARPC1B and PTEN, and ARPC1B and ERG. Knockdown of ARPC1B impaired the migration and invasion of PC3 and DU145 PCa cells via downregulation of Aurora A kinase (AURKA) and resulted in the arrest of the cells in the G2/M checkpoint of the cell cycle. Additionally, higher ARPC1B expression was observed in stable PC3-ERG cells compared to normal PC3, supporting the association between ERG and ARPC1B. Our findings implicate the role of ARPC1B in PCa invasion and metastasis in association with ERG and further support its prognostic value as a biomarker in association with ERG and PTEN in identifying aggressive phenotypes of PCa cancer.

## 1. Introduction

Prostate cancer (PCa) is, worldwide, the most common type of cancer among men [1]. Therefore, identifying molecular biomarkers related to PCa progression and patient prognosis remains an important topic in cancer research. To date, the only two promising biomarkers that are implemented clinically are the use of ERG and PTEN expression, which can be assessed by immunohistochemistry (IHC) in routine clinical pathology laboratories. 

The actin-related protein 2/3 complex subunit 1B(ARPC1B) gene is one of the seven actin cytoskeleton subunits of the human Arp2/3 protein complex [2]. The Arp2/3 protein complex includes the actin-related ARPC1, which contains the two ARPC1A and ARPC1B isoforms, ARP2, ARP3, ARPC2, ARPC3, ARPC4, and ARPC5 [3]. This complex appears to be an essential part of most eukaryotic cells because it directly regulates cell motility [4]. The ARPC1B protein is remarkably similar to the protein encoded by the ARPC1A gene. Their similarity implies that together they may function as the p41 subunit in the Arp2/3 complex by assembling the structure of this complex. Further, this similarity may imply that ARPC1B plays a key role in the branching of actin filaments [5]. 

Despite this, the role and functions of ARPC1B in cancer are not fully understood. ARPC1B is mostly responsible for regulating the nucleation and assembly of actin monomers into microfilaments [6]. Actin filaments play a major role in the formation of the cytoskeleton of cells, cell–cell junctions, and motility of many pathogens [7,8]. Further, it has been shown that actin nucleation is directly related to the production of pseudopodia in cancer cells, which is critical to the ability of cancer cells to migrate and invade [9]. The ARPC1B protein has also been implicated in lamellipodia sheet membrane formation at the edge of motile cells, which relates directly to cancer cell migration and metastasis in osseous and soft tissue [3,10]. The overexpression of the Arpc2/3 members in hepatocellular carcinoma was shown to be significantly correlated with worse overall survival and shorter progression-free survival [11].

Because ARPC1B is one of the main components of the Arp2/3 complex, there is a considerable possibility that ARPC1B is directly involved in cell migration and can be studied further as a promising biomarker for aggressive PCa. 

## 2. Materials and Methods

### 2.1. Study Population and Tissue Microarray Construction

To assess ARPC1B association to disease progression, we utilized tissue microarray (TMA) constructed from a cohort of 160 patients diagnosed with prostate cancer. The cases included castration resistant prostate cancer (CRPC) cores (*n* = 60), localized prostate cancer cores (*n* = 157) and benign cases cores (*n* = 83). 

To assess the association of ARPC1B to patient’s prognosis, we assessed a second cohort of men diagnosed with prostate cancer via transurethral resection of the prostate (TURP) (*n* = 296). Patients within the TURP cohorts were either not actively treated or treated by androgen deprivation therapy (ADT) with a luteinizing hormone-releasing hormone (LHRH) agonist for disease progression post TURP (advanced group) or prior to the TURP (patients with a previous diagnosis of PCa), characterized as castrate-resistant prostate cancer (CRPC). 

Follow-up information was collected from the Alberta Tumor Registry for overall survival and cancer specific mortality. The samples from the TURP cohort were assembled onto 2 TMAs with an average two cores per patient using a manual tissue arrayer (Beecher Instruments, Silver Spring, MD, USA). 

The study was approved by the University of Calgary Cumming School of Medicine Ethics Review Board and in accordance with the 1964 Helsinki declaration and its later amendments and comparable ethical standards. 

### 2.2. Pathological Analysis

In each cohort, histological diagnoses of individual TMA cores were confirmed by the study pathologists (RS and TAB) on the initial slides. Gleason scoring was assessed according to the 2018 WHO and ISUP grade groups. In each patient, the predominant two patterns of PCa were sampled and included on the TMAs for analysis. ARPC1B intensity was evaluated on 4-tiered system (0–3 corresponding to negative to high intensity). To confirm reliability of the two cohorts, several pathological parameters were confirmed to reflect prognostic value such as surgical margins, and Pt and Gleason scores. Figure 1 shows examples of variable intensity of ARPC1B in different prostate cancer tissue types. ARPC1B expression scores were also assessed in combination to PTEN and ERG intensity and grouped as combined signatures of high-risk vs. low risk. The high-risk groups combined ARPC1B intensity (weak, moderate, and high) and PTEN negative or ERG positive.

### 2.3. Immunohistochemistry

The IHC stain was performed on Dako Omnis auto-stainer (Agilent, Santa Clara, CA, USA) at the Alberta Precision Research Lab of Calgary Laboratory Services as routine procedure. Briefly, formalin-fixed paraffin-embedded (FFEP) sections were dewaxed and hydrated in descending grades of ethanol and water and then pre-treated with an antigen retrieval buffer, followed by incubation with ARPC1B (1:200) Rabbit polyclonal (HPA004832, Sigma, St. Louis, MO, USA) primary antibody. Then the samples were washed and incubated with HRP secondary antibody. The signal was developed using DAB+ Substrate Chromogen system (Agilent, Santa Clara, CA, USA ). ARPC1B expression was assessed using a 4-tiered system (negative—0; weak—1; moderate—2; high expression—3). 

PTEN and ERG IHC were evaluated as binary values (negative vs. positive) reflective of ERG gene rearrangements and homozygous PTEN deletions as detected by FISH, as previously described [12,13]. 

### 2.4. Cell Lines

The human prostate cancer cell lines LNCaP, VcaP, PC3, C4-2, and DU145 were used in this project. All cell lines were purchased from American Type Culture Collection (ATCC; Manassas, CA, USA). The RWPE-1 cell lines were used as a positive control in the Western blot experiments. 

PC3 prostate cancer lines were cultured in DMEM/F12 (GIBCO life technology, Grand Island, NY, USA), and LNCaP prostate cancer lines were cultured in RPMI 1640 medium (GIBCO life technology, Grand Island, NY, USA). DU145, C4-2, and VcaP cell lines were grown in DMEM media (GIBCO life technology, Grand Island, NY, USA) supplemented with 10% FBS (GIBCO life technology, Grand Island, NY, USA) at 37 °C in 5% CO_2_ atmosphere. RWPE-1 cells were grown in Keratinocyte-Serum Free Medium (GIBCO life technology, Grand Island, NY, USA) Stable PC3-ERG cell lines were obtained from Felix Feng, University of Michigan [14].

### 2.5. Cell Line Transfection and RNA Silencing

ARPC1B knockdown was performed using an ARPC1B Silencer select pre-designed siRNA and a scrambled siRNA used as a negative control (Ambion, Grand Island, NY, USA). Briefly, PC3 and DU145 or PC3-ERG cells were seeded in six well plates until 70–80% confluency was reached. Then the siRNA transfection reaction mix was prepared with opti-MEM (GIBCO life technology, Grand Island, NY, USA) and Lipofectamine RNAiMAX (Invitrogen, Carlsbad, CA, USA) according to the manufacturer’s instructions. Then Western blot was performed to determine the efficiency and duration of the ARPC1B knock down.

### 2.6. Proliferation Assay (MTS)

The cells viability was assessed using proliferation assay. PC3 cells were seeded for 48 h in a 96 well plate. Cell number and viability was determined using CellTiter-Aqueous MTS assay (Promega, CA, USA) and then the media was changed to opti-MEM (GIBCO life technology, Grand Island, NY, USA). Then the cells were transfected with ARPC1B siRNA1, siRNA2, or control scramble siRNA as the negative control. They were incubated in humidified tissues culture incubator with 5% CO_2_ atmosphere for different time intervals at 37 °C. Following this, MTS and PMS (Promega, CA, USA) solution was added to each well according to manufacturer instruction. The plate was incubated again for 2 h and the absorbance reading was measured at 490 nm using an ELISA plate reader. Each experiment was repeated 5 times. 

### 2.7. Wound Healing Assay

The PC3 cell line was seeded in 4 Well Insert (Ibidi, Cat # 80469) with DMEM/F12 medium until 80–90% confluency was reached. Then cells were transfected with ARPC1B siRNA1, siRNA2, or the scrambled siRNA as a negative control to compare the difference between the migrated cells. The wound healing of each group was observed at 0-, 24-, and 48-h post transfection. Cells were captured using a phase contrast inverted EVOS FL life microscope with phase contrast 10× magnification.

### 2.8. Migration Invasion Assay

The PC3 and DU145 cell lines were seeded in 6 well plates, and were further transfected with ARPC1B siRNA1, siRNA2, or the scrambled siRNA as a negative control. A total of 24 h post transfection, the cells were trypsinized and placed on the top of BD Biocoat control inserts for migration assay and Matrigel for invasion assay (BD Biosciences, San Jose, CA, USA). After 48 h, the cells were fixed and stained with Diff Quick (Siemens Healthcare diagnostics, Tarrytown, NY, USA). Cells were captured using an inverted EVOS FL life microscope with brightfield 10× and 40× magnification. The number of cells for multiple frames were counted, averaged from 40× magnification, and compared to the negative control.

### 2.9. Flow Cytometry

For Annexin V/PI, assay cells were prepared as described in MM Section 2.5, then trypsinated and treated with Dead Cell Apoptosis Kits with Annexin V for Flow Cytometry Ca# V13241 (Invitrogen, Carlsbad, CA, USA), following manufacturer’s instructions. Then the results were analyzed using BD LSR II Flow Cytometer. For cell cycle analysis, ARPC1B knockdown and correspondent controls with appropriate replications were prepared as previously described for knockdown. They were further harvested, washed in cold PBS, fixed in 70% ethanol, and stained with 50 µg/mL propidium iodide and 100 µg/mL Rnase A in PBS. The cells were analyzed for their DNA content with a BD LSR II Flow Cytometer. The data were further analyzed using FlowJo™ v10 Software-BD Biosciences.

### 2.10. Western Blot

Total protein was extracted using RIPA buffer (Sigma–Aldrich, St. Louis, MO, USA) with cocktail protease inhibitors and PMSF (Roche, Rotkreuz, Switzerland). Equal quantities of proteins were loaded into each lane and separated on polyacrylamide SDS gel. Then the proteins were transferred to the PVDF or nitro cellulose membrane (BIO-RAD Immun-Blot^®^ Membrane) and the membrane was further incubated and placed on a shaker with blocking buffer containing 10% skimmed milk in TBS for 1 hr at room temperature. Then they were incubated with primary antibody (Appendix A) for 1 h at 37 °C or overnight in 4 °C with consistent shaking. This was followed by incubation with either anti-rabbit IgG or anti-mouse IgG secondary antibody conjugated to HRP horseradish peroxidase (Cell Signaling, Danvers, MA, USA) in blocking buffer for 1 hr at room temperature. The ECL substrate Chemiluminescence signal was detected using ChemiDoc imaging system (Bio-Rad Laboratories, Hercules, CA, USA). 

## 3. Statistical Analysis

Descriptive statistics were used to describe the study data. Frequency and proportions were reported for categorical data. Mean and standard deviations were reported for continuous data. Chi-square tests were used to compare two categorical variables, Fisher’s exact test was used for cell frequency lower than five. The overall survival (OS) was defined as the time from diagnosis to death; subjects who were alive at the end of the study period were censored. Prostate cancer specific mortality (PCSM) was defined as death due to prostate cancer, subjects who died due to other reason or who were alive at the end of the study period were censored. Relapse free survival (RFS) was calculated from the date of treatment to the date of relapse. OS, PCSM, and RFS were analyzed using Kaplan–Meier method. Median time and the 95% confidence interval were reported. Log rank tests were used to compare two or more survival curves. Cox’s proportional hazard model was used to determine the hazard ratio (HR) and the corresponding 95% confidence interval. Adjusted Cox’s model was fitted as well. All statistical analyses were conducted using IBM SPSS version 23. A *p*-value < 0.05 was used for statistical significance and two-sided t tests were used. 

## 4. Results

### 4.1. ARPC1B Expression in Prostate Cancer Progression

Overall, there was an increase in expression of the ARPC1B in castration-resistant prostate cancer (CRPC), and localized PCa compared with benign prostate tissue through the IHC Figure 1. The mean intensity values were significant between benign, localized, and CRPC PCa samples (*p* < 0.01). Mean intensity values were all weak and under one, and CRPC and localized PCa showed close intensities but were significantly higher in benign prostate tissue (*p* < 0.01). The means intensity of ARPC1B expression was 0.06 ± 0.044, 0.26 ± 0.05, and 0.27 ± 0.07 in the benign, localized PCa, and CRPC, respectively (values expressed in mean ± SEM) (Figure 1).

### 4.2. ARPC1B Expression in Relation to Gleason Grade Grouping, and ERG and PTEN Protein Expression

High ARPC1B expression was noted in 47% of GG5 cases vs. 29% of GG3. Additionally, high ARPC1B intensity was noted in 37% of PTEN negative cases and 23% of ERG positive cases, but this was not statistically significant.

### 4.3. ARPC1B Expression in Relation to Overall Survival (OS) and Cause-Specific Survival (CSS)

In this cohort, higher ARPC1B intensity was only marginally associated with OS, but not with cause-specific mortality (HR 1.28, CI: 0.95–1.72, *p* = 0.10) and HR 1.2, CI 0.79–1.83, *p* = 0.39), respectively. 

Because PTEN and ERG are well known to be associated with PCa patient’s prognosis, we sought to investigate the prognostic value of combined ARPC1B with PTEN or ERG. PTEN negative combined with ARPC1B expression was associated with poor overall survival (HR: 2.91, CI: 1.92–4.43; *p* < 0.0001) and cause-specific mortality (HR3.53, CI: 2.04–6.09; *p* < 0.0001) (Figure 2A,B). This was more significant compared to PTEN negative alone for OS and CCS (HR 2.22, CI: 165–2.99; *p* < 0.0001) and (HR 2.96, CI: 1.95–4.48, *p* < 0.0001), respectively (Figure 2C,D). 

Combining ARPC1B with ERG positivity showed similar results to OS and CCS (HR 2.02, CI: 1.26–3.23; *p* = 0.003) and (HR 2.05, CI: 1.11–3.81; *p* = 0.02), respectively (Figure 2E,F). This was also more significant to ERG alone (data not shown). However, only ARCB1P in association to PTEN remained statistically significant when adjusting to Gleason grade groups for OS and marginally for CCS (Table 1). 

Additionally, we confirmed the prognostic significance of ARPC1B genomic alteration using all cohorts in the TCGA public database for overall prostate cancer Figure 3A and bladder cancer disease progression Figure 3B (9041 patients/9353 samples in 24 studies for prostate and 2678 patients/2769 samples in 17 studies for bladder).

### 4.4. ARPC1B Expression in the Prostate Cancer Cell Lines

ARPC1B protein expression levels were investigated using Western blot. High expression levels were observed in PC3 and DU145 cell lines as well as in the RWPE-1 immortalized prostate cell line. In comparison, the C4-2, LNCaP, and VCaP cell lines show relatively low ARPC1B expression, as shown in Figure 4A. 

### 4.5. Knockdown of ARPC1B in PC3 and DU145 Prostate Cancer Cells

Based on the above observation, we used PC3 and DU145 cell lines to assess the ARPC1B functions in prostate cancer cell lines. ARPC1B knockdown was successfully obtained using siRNA in PC3 and DU145 cell lines Figure 4B. The optimal duration of knockdown was also observed at 48 h and lasted up to 96 h. Because our clinical data indicated potential correlation of OS and CCS in patients with ERG gain, we investigated ARPC1B expression in PC3 cells that were stably overexpressing ERG (PC3-ERG cells.) Our results indicated a significant upregulation of ARPC1B in the PC3-ERG cell line compared to PC3 parental cells. ERG protein expression was validated in PC3-ERG overexpressed cells, as shown in Figure 3. Additionally, ARPC1B was knocked down in PC3-ERG cells, as shown in Figure 4D.

### 4.6. Knockdown of ARPC1B Reduces Prostate Cancer Cell Proliferation, Migration, and Invasion

To investigate the potential function of ARPC1B in tumorigenesis in vitro, we performed cells proliferation assay, wound healing assay, and transwell cell migration and invasion assay. ARPC1B knockdown significantly attenuates PC3 cells proliferation, as shown in Figure 5A, and reduced the wound healing ability of these high metastatic cells, as shown in Figure 5B. Furthermore, silencing ARPC1B significantly impaired PC3 and DU145 prostate cancer cells migration and invasion ability, as shown in Figure 6A. 

### 4.7. ARPC1B Regulates PCa Cell Cycle through Aurora A Kinase (AURKA)

We investigated the possible link of ARPC1B to induce PCa apoptosis. Our data indicated that there was no significant difference in apoptosis using flowcytometry after the knockdown of ARPC1B (Appendix A). However, cell cycle analysis showed that ARPC1B knockdown causes a delay of cells to enter G2/M and, therefore, causes them to be arrested at the G2/M phase (Figure 6B). We further confirmed the protein expression level of the cell cycle regulator Aurora A kinase (AURKA) in PC3 and PC3-ERG cells. Our results confirmed significant down-regulation of AURKA, Lamin A, and Cyclin B1 in PC3 cells and PC3-ERG (Figure 6C). However, the degree of downregulation of those markers was lower in PC3-ERG compared to PC3 cells. These results suggest a critical role of ARPC1B modulation of PCa cell cycle.

## 5. Discussion

In this study, we investigated the clinical significance of ARPC1B protein and its role in PCa lethal disease prognosis. Our findings indicated that high expression of ARPC1B increases with disease progression, and it correlates with a poor overall survival and specific-cause mortality. This prognostic significance was further amplified when ARPC1B expression was combined with PTEN loss or ERG gain. 

PTEN and ERG are well known to be associated with PCa patient’s prognosis [15] and therefore, we sought to investigate the prognostic value of combined ARPC1B with PTEN or ERG. PTEN negative or ERG positive combined with ARPC1B expression was associated with poor overall survival. However, only ARPC1B expression in association to PTEN remained statistically significant when adjusting to Gleason grade groups for OS and marginally for CCS (Table 1). Additionally, we confirmed the prognostic significance of ARPC1B genomic alteration using two cohorts in the TCGA public database for overall prostate cancer and bladder cancer disease progression. Several studies have reported the association between ERG expression and favorable or unfavorable outcomes in PCa [16,17]. On the other hand, PTEN loss has generally been linked to unfavorable outcomes in PCa, as shown in our results as well [18]. 

To investigate and validate the role of ARPC1B in PCa, we screened the protein expression of several cell lines in vitro. Previous studies suggest that ARP2/3 subunits are overexpressed in human cancer [19,20,21,22,23,24]. We established that the knockdown of ARPC1B impaired the migration and invasion ability of PCa cells in vitro. In addition, previous studies have demonstrated that the knockdown of ARP2/3 complex in mouse embryonic fibroblasts directly inhibits the lamellipodia formation and therefore influences the directionality of cell movement, making mouse embryonic fibroblasts unable to sense changes in the extracellular matrix, suppressing their ability to move along these gradients in a coherent manner [11].

Interestingly, our data indicated that the ARPC1B knockdown in PCa cells delayed the entry of most cells into the mitotic phase. This has been suggested to be the major role of ARPC1B, which mediates the transition from S to the G2/M phase in PCa cell cycle, presumably due to its inherent role in the activation of AURKA [20,25]. It has been demonstrated previously that ARPC1B function as an activator and substrate for AURKA [20]. In addition, our data show that the knockdown of ARPC1B inhibits AURKA activity in cells, which, in turn, diminishes the ability of cells to enter mitosis [15]. AURKA plays an important role in the regulation of spindle formation and mitosis [20]. Interestingly, the expression of AURKA has been documented in the androgen-regulated CRPC cells [26], and several studies have demonstrated the role of AURKA as a potential therapeutic target for PCa [27]. In addition, inhibition of AURKA has been shown to potentially benefit patients with Neuroendocrine Prostate Cancer [28]. It is thought that the coordinated action of the AURKA and the cofactor Bora, activate PLK1 as cells approach the M-phase [20]. The G2/M DNA damage checkpoint serves to prevent the cell from entering mitosis (M-phase), causing genomic DNA damage. Specifically, the activity of the Cyclin B-cdc2 (CDK1) complex is pivotal in regulating the G2/M-phase transition. We demonstrated that after the knockdown of ARPC1B in PCa cell lines, the expression of Cyclin B1 was significantly decreased. This in turn points to the critical role of ARPC1B in regulating the G2/M-phase transition and its role as a cell cycle regulator. Of note, the level of Cyclin B1 and N-Cadherin was not as striking in PC3-ERG compared to PC3 cells when ARPC1B was knocked down, suggesting the potential role of ERG in invasion and metastasis in PCa. In a previous study, ERG upregulation is seen in about half of all prostate cancer patients, and combining human prostate cancer genetics with transgenic mice showed that PTEN loss occurs concomitantly with ERG aberrant expression [29,30]. Therefore, the role of ARPC1B and its correlation with ERG gain and PTEN loss can have an association on cell migration and invasion, which may promote the progression of invasive prostate cancer.

Our results propose that further in-depth investigation is needed to identify the mechanism underlining the potential role of ARPC1B in the tumorigenesis of PCa cells. Our findings suggest that ARPC1B functions as a complex cell cycle regulator with AURKA and it may be involving in invasion and migration of PCa cells. Our findings create new possibilities for the role of ARPC1B as a potential biomarker for lethal aggressive PCa. 

## Figures and Tables

**Figure 1 ijms-23-01476-f001:**
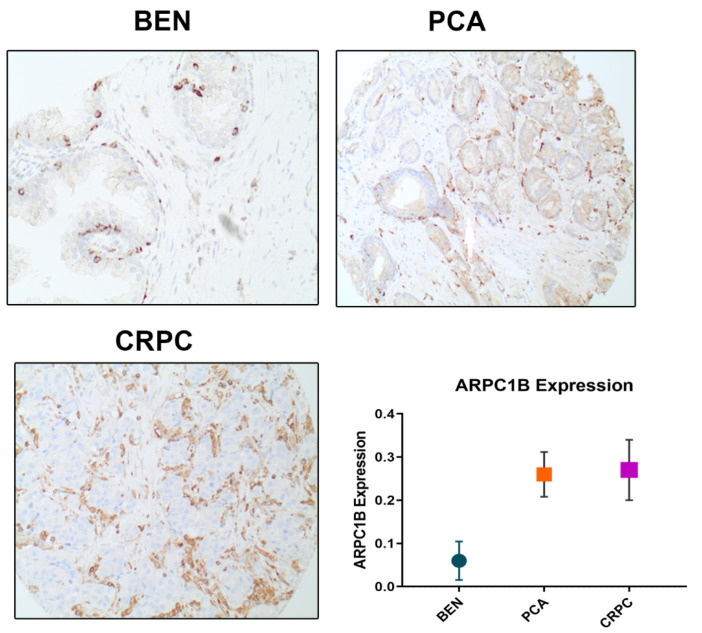
ARPC1B expression in clinical prostate cancer cases. Immunohistochemistry staining (IHC) shows ARPC1B expression in Benign (BEN, 10×), localized prostate cancer (PCA, 4×), and castration resistance prostate cancer (CRPC, 10×) in human tissue samples. The mean expression of ARPC1B in BEN, PCA, and CRPC in human tissue samples. ARPC1B protein expression levels were scored through IHC. Each sample was scored semi quantitatively using four-tiered system (negative—0; weak—1; moderate—2; strong—3). The error bars indicate the Standard error of the mean. Student t-test was performed, *p* value < 0.05 was considered significant between prostate cancer CRPC, and localized PCa was compared with benign. *p* < 0.0144, *p* < 0.0123.

**Figure 2 ijms-23-01476-f002:**
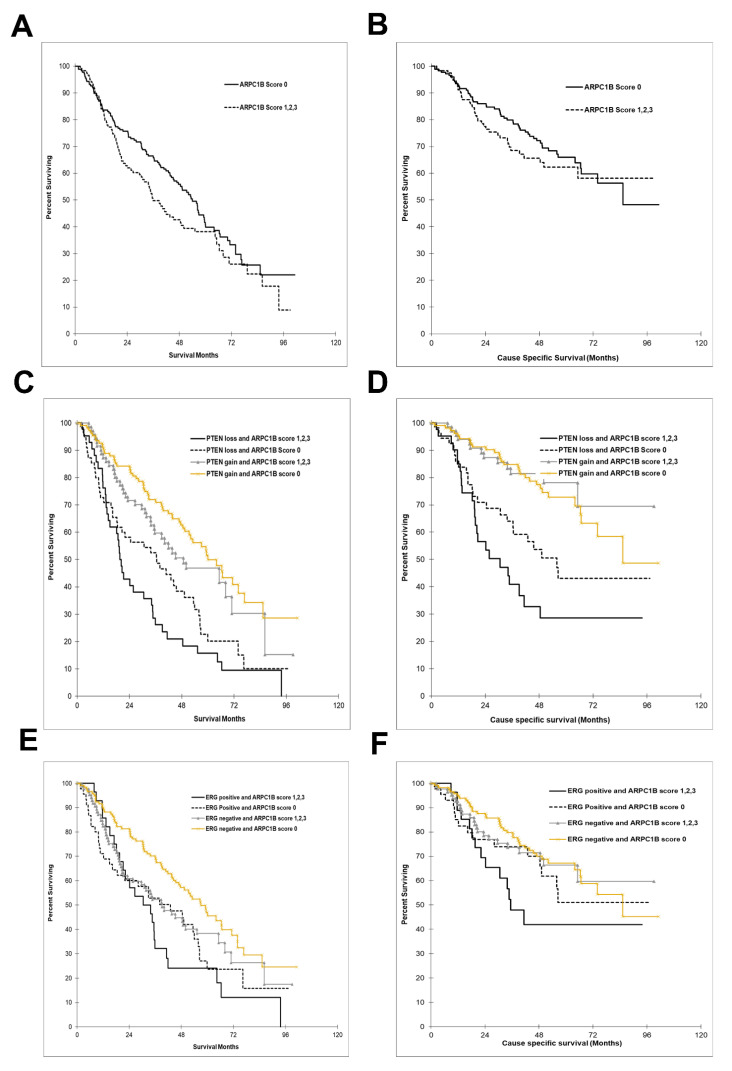
Kaplan–Meier (KM) curves demonstrating ARPC1B expression in relation to prostate cancer cases overall survival and cause-specific survival. (**A**) overall survival according to ARPC1B IHC score of and (**B**) KM curves for cause-specific survival with ARPC1B score (**C**,**D**) ARPC1B and PTEN loss. (**E**,**F**) ARPC1B and ERG gain. Each sample was scored semi-quantitatively for ARPC1B using four-tiered system (negative—0; weak—1; moderate—2; strong—3). PTEN loss = PTEN negative staining. PTEN gain = weak, moderate, or high staining.

**Figure 3 ijms-23-01476-f003:**
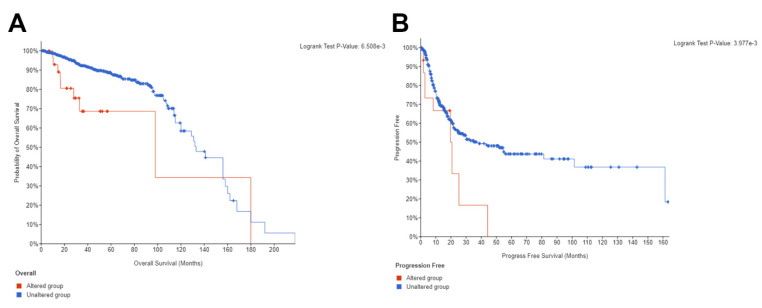
ARPC1B genomic alteration KM for (**A**) PCa overall survival and (**B**) Bladder cancer progression free survival. Data obtained from cBioPortal (contains all TCGA datasets, ~12.5 K samples combined for the bladder and prostate cancer together.

**Figure 4 ijms-23-01476-f004:**
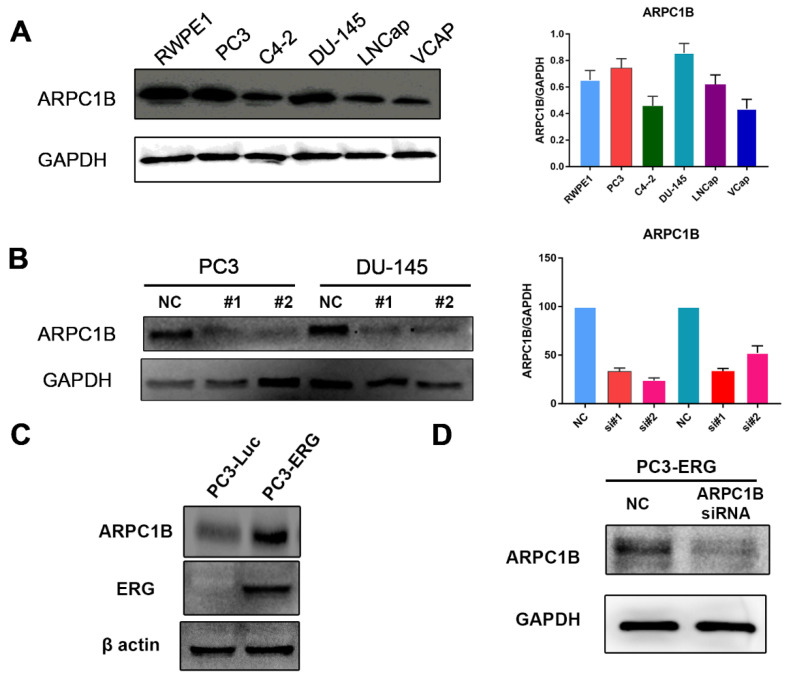
Expression of ARPC1B in PCa cell lines. (**A**) Western blot analysis of ARPC1B expression in PCa cell lines: RWPE1, PC3, C4-2, DU145, LNCaP, and VCAP. (**B**) Knockdown of PC3 and DU145 cell lines using ARPC1B siRNA#1, siRNA#2, or control scramble siRNA as the negative control. (**C**) The expression of ARPC1B in PC3-ERG overexpression cells (**D**) Western blot analysis of ARPC1B knockdown in PC3-ERG cells using ARPC1B, siRNA#1, or scramble siRNA as the negative control.

**Figure 5 ijms-23-01476-f005:**
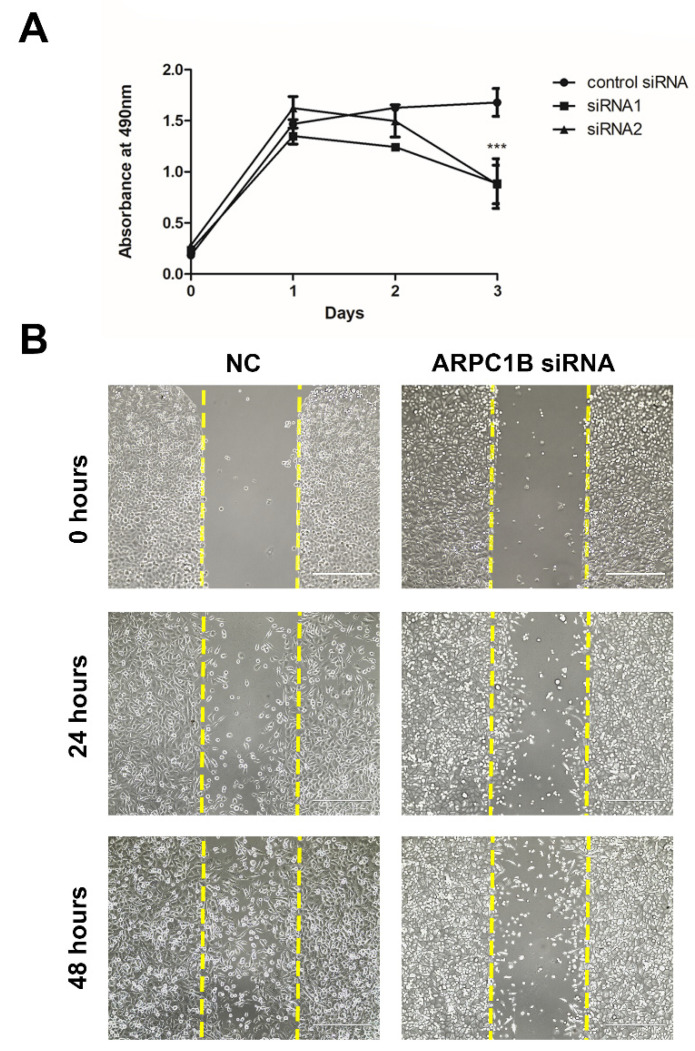
Knockdown of ARPC1B in PC3 cell line attenuates PCa cell proliferation and migration. (**A**) Relative PC3 cells proliferation using MTS assay over three days. Statistical significance calculated using, two-way ANOVA, *** shows *p* < 0.001. Error bars indicate the SEM. (**B**) wound healing assay for PC3 cells after ARPC1B knockdown at 0, 24, and 48 h.

**Figure 6 ijms-23-01476-f006:**
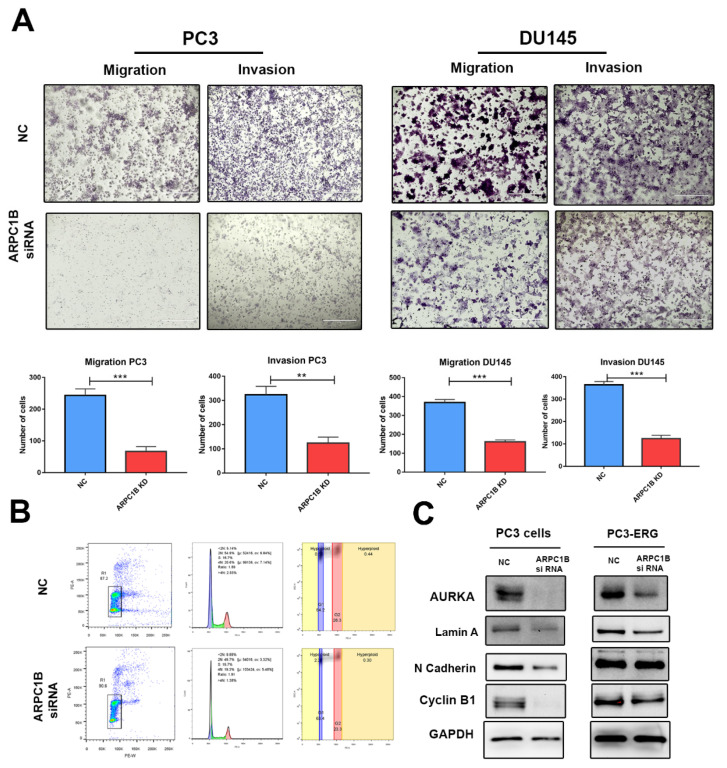
ARPC1 knockdown attenuates PCa cell migration and invasion. (**A**) Migration and invasion assay for PC3 and DU145 cell lines after 48 h of ARPC1B knockdown. *** *p* < 0.001, ** *p* < 0.01. Scale bar 400 µM. (**B**) Cell cycle analysis using PI staining. Data analyzed and presented using FlowJo. (**C**) Western blot analysis of AURKA, Lamin A, N-Cadherin, Cyclin B1 in PC3, and PC3-ERG cell transfected with either ARPC1B, siRNA, or control scramble siRNA as the negative control. GAPDH used as loading control.

**Table 1 ijms-23-01476-t001:** Univariate and multivariate analysis and HR of the different biomarker combinations to overall survival and cancer-specific mortality in the TURP cohort.

Variables	Overall Survival		Prostate Cancer-Specific Survival	
	HR (95% CI)	*p*-Value	HR (95% CI)	*p*-Value
PTEN (Positive—scores 1,2,3)				
Negative—score 0	2.22 (1.65–2.99)	<0.0001	2.96 (1.95–4.48)	<0.0001
ARPC1B (score of 0)				
Scores 1,2,3	1.28 (0.95–1.72)	0.102	1.20 0.79–1.83)	0.393
Combination of PTEN and ARPC1B (PTEN 1–3 and ARPC1B score = 0)				
PTEN 0 and ARPC1B scores 1,2,3	2.91 (1.92–4.43)	<0.0001	3.53 (2.04–6.09)	<0.0001
PTEN 0 and ARPC1B score = 0	2.14 (1.44–3.20)	<0.0001	2.26 (1.31–3.89)	0.003
PTEN 1–3 and ARPC1B scores 1,2,3	1.28 (0.85–1.93)	0.238	0.80 (0.42–1.55)	0.803
Combination of PTEN and ARPC1B (PTEN 1–3 and ARPC1B score = 0) *				
PTEN 0 and ARPC1B scores 1,2,3	1.96 (1.24–3.10)	0.004	1.69 (0.93–3.07)	0.086
PTEN 1–3 and ARPC1B score = 0	1.53 (0.99–2.35)	0.053	1.23 (0.69–2.21)	0.487
PTEN 0 and ARPC1B scores 1,2,3	1.21 (0.79–1.86)	0.374	0.69 (0.35–1.36)	0.283
Combination of ERG and ARPC1B (ERG negative and ARC1PB score = 0)				
ERG positive and ARPC1B scores 1,2,3	2.02 (1.26–3.23)	0.003	2.05 (1.11 −3.81)	0.022
ERG positive and ARPC1B score 0	1.69 (1.10–2.57)	0.016	1.37 (0.75–2.49)	0.300
ERG negative and ARPC1B scores 1,2,3	1.41 (0.98–2.03)	0.062	1.06 (0.63–1.77)	0.830
Combination of ERG and ARPC1B (ERG negative and ARC1PB score = 0) *				
ERG positive and ARPC1B scores 1,2,3	1.44 (0.87–2.38)	0.157	1.09 (0.56–2.14)	0.797
ERG positive and ARPC1B score 0	1.23 (0.79–1.93)	0.357	0.75 (0.39–1.43)	0.386
ERG negative and ARPC1B scores 1,2,3	1.29 (0.88–1.89)	0.191	0.83 (0.48–1.44)	0.515

* Adjusted for Gleason score. ARPC1B, PTEN scores; negative—0; weak—1; moderate—2; high—3.

## Data Availability

Not applicable.

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
