# Peer review of "ARPC1B Is Associated with Lethal Prostate Cancer and Its Inhibition Decreases Cell Invasion and Migration In Vitro"

_ijms, 2022, doi:10.3390/ijms23031476_

Round 1

Reviewer 1 Report

Gamallat et al. reported in this manuscript that ARPC1B (Actin Related Protein 2/3 Complex Subunit 1B) is associated with poor prognosis of prostate cancer, and is involved in promoting metastasis and invasion. The authors provided strong clinical evidence and linked the expression of ARPC1B with ERG, the oncogene for prostate cancer. However, some concerns for the study might need to be answered. 

  1. While expression of ARPC1B can be regulated by ERG. The authors should provide or discuss the potential mechanism.
  2. If high expression of ARPC1B  were critical for progression of PCs, the experiments for gain of functions, such as ARPC1B-overexpression, might be important than experiments with ARPC1B knockdown shown in the manuscript.
  3. The association of PTEN loss with ARPC1B has been addressed with clinical samples (Fig. 1), but never further evaluated. Is PTEN really involved in  the regulation of ARPC1B?

Minor suggestion:

The format of subtitles in the result section should be checked. 

Author Response

Reviewer 1,

Gamallat et al. reported in this manuscript that ARPC1B (Actin Related Protein 2/3 Complex Subunit 1B) is associated with poor prognosis of prostate cancer, and is involved in promoting metastasis and invasion. The authors provided strong clinical evidence and linked the expression of ARPC1B with ERG, the oncogene for prostate cancer. However, some concerns for the study might need to be answered. 

  1. While expression of ARPC1B can be regulated by ERG. The authors should provide or discuss the potential mechanism.

#The potential mechanism has been discussed as recommended.

  1. If high expression of ARPC1B were critical for progression of PCs, the experiments for gain of functions, such as ARPC1B-overexpression, might be important than experiments with ARPC1B knockdown shown in the manuscript.

#We thank the reviewer for his comment. Since we did observe increased ACPC1B expression in PCA samples vs Benign, we aimed to investigate whether decreasing ARPC1B would be associated with decrease invasion and metastasis suggesting potential therapeutic targets.

  1. The association of PTEN loss with ARPC1B has been addressed with clinical samples (Fig. 1), but never further evaluated. Is PTEN really involved in the regulation of ARPC1B?

#To investigate this, it would require significant mechanistic and invitro studies which is beyond the scope of the current study. We will take the reviewers comments and plan future studies in due course.

Minor suggestion:

The format of subtitles in the result section should be checked. 

Thank you for your feedback. The format of subtitles has been checked.

Reviewer 2 Report

In this work, the authors first used IHC analysis on arrayed prostate tissues to detect the expression status of ARCP1B in benign, localised and castration-resistant prostate cancer samples, then they studied the effect of ARCP1B on cell migration and invasion with two highly malignant prostate cancer cell lines. The authors concluded that ARCP1B is a biomarker and a tumour promoter for prostate cancer. In addition, the authors also performed some more experiment to elucidate the molecular mechanisms and suggested that ARCP1B may promote the malignant progression via upregulation of AURKA Kinase A (AURKA).

This manuscript presented some valuable data and the work is interesting. Generally speaking the experiments were conducted well and the statistical analysis was correct. The presentation is relatively clear. There are 2 concerns for this work, mainly on the ARCP1B functional characterisation part:

  1. The current results are not sufficient to enable the authors to claim ARCP1B is either an independent diagnostic or prognostic marker. Although the overall the level of ARCP1B is significantly higher in carcinomas, there is no difference between carcinomas with different degree of malignancy, and only a proportion of carcinomas express higher level than the benign. In another words, many carcinomas express similar levels of ARCP1B to the benign, thus if it is used as a diagnostic marker, it would have a large number of false positive. If it is used as prognostic marker, the author should compare it with the currently used markers: combined Gleason scores; PSA (although PSA is not good for diagnosis, it is a very good marker for clinical management), and AR-index. Authors should show evidence that ARCP1B at least is as good as these currently used markers. To relate this with EGR is not appropriate in this regard, since EGR’s role in prostate cancer is still uncertain. PTEN tumour suppressor may be a potential prognostic marker. If the authors want to claim that ARCP1B is related to PTEN in this regard, they should first detect the expression status of PTEN in the same samples, then compare ARCP1B and PTEN separately as markers, then use multivariate test to assess how more accurate when both jointly used than PTEN singly used for predicting the patient outcomes.
  2. The authors claimed the “suppression of ARCP1B decreases cell proliferation and metastasis in-vitro”. In fact, there is no data on cell proliferation assay. Migration and invasion assays can only test certain malignant characteristics, they can not replace the metastatic process test. Due to the fact that ARCP1B is not implicated in prostate cancer previously, the authors should conduct proliferation assay, wound healing assay and the anchorage-independent growth (as an indication of tumorigenicity) assay. The animal assay should also be conducted.

Finally, from the history of p53 study, when mutation occurs, tumour-suppressor could be mistakenly thought as a promoter. For a new gene like ARCP1B, the authors may want to ensure that there are no mutations occurred.

Author Response

Reviewer 2,

  1. The current results are not sufficient to enable the authors to claim ARCP1B is either an independent diagnostic or prognostic marker. Although the overall the level of ARCP1B is significantly higher in carcinomas, there is no difference between carcinomas with different degree of malignancy, and only a proportion of carcinomas express higher level than the benign. In another words, many carcinomas express similar levels of ARCP1B to the benign, thus if it is used as a diagnostic marker, it would have a large number of false positive. If it is used as prognostic marker, the author should compare it with the currently used markers: combined Gleason scores; PSA (although PSA is not good for diagnosis, it is a very good marker for clinical management), and AR-index. Authors should show evidence that ARCP1B at least is as good as these currently used markers. To relate this with EGR is not appropriate in this regard, since EGR’s role in prostate cancer is still uncertain. PTEN tumour suppressor may be a potential prognostic marker. If the authors want to claim that ARCP1B is related to PTEN in this regard, they should first detect the expression status of PTEN in the same samples, then compare ARCP1B and PTEN separately as markers, then use multivariate test to assess how more accurate when both jointly used than PTEN singly used for predicting the patient outcomes.

#There was no significant association in tissue samples PTEN and ARPC1B expression. However, combining the two markers in a biomarkers panel showed significant prognostic value related to overall and cause specific survival. This was still present when adjusting to Gleason score in multivariate analysis. The data are shown in Table 1 with the combined PTEN and ARPC1B as univariate and multivariate (adjusting for Gleason score). In addition to that we have also included univariate of ERG, and multivariate of PTEN alone, ARPC1B alone and ERG alone adjusting for GS

  1. The authors claimed the “suppression of ARCP1B decreases cell proliferation and metastasis in-vitro”. In fact, there is no data on cell proliferation assay. Migration and invasion assays can only test certain malignant characteristics, they can not replace the metastatic process test. Due to the fact that ARCP1B is not implicated in prostate cancer previously, the authors should conduct proliferation assay, wound healing assay and the anchorage-independent growth (as an indication of tumorigenicity) assay. The animal assay should also be conducted.

# Thank you for your suggestion. We changed the title to only include invasion and migration instead of cell proliferation assay to be more in line with our findings.

Finally, from the history of p53 study, when mutation occurs, tumour-suppressor could be mistakenly thought as a promoter. For a new gene like ARCP1B, the authors may want to ensure that there are no mutations occurred

# ACPC1B gene mutation has been reported to cause immune deficiency and thrombocytopenia. In C-Bioportal, mutations or this gene is rare and not reported in prostate cancer. Mostly is it amplification of 1.4% and deep deletion of 0.2%. There is no reports on this gene being mutated in prostate cancer and causing increase in protein expression as in p53.

Round 2

Reviewer 2 Report

The authors have addressed all issues raised and changed version is now ready for publication.
